# Sexual orientation predicts men's preferences for sexually dimorphic face-shape characteristics: A replication study

**Victor Shiramizu** ⬛ *, **Ciaran Docherty, Lisa M. DeBruine, Benedict C. Jones**

Institute of Neuroscience & Psychology, University of Glasgow, United Kingdom

* victor.shiramizu@glasgow.ac.uk

## Abstract

Many researchers have proposed that straight men prefer women's faces displaying feminine shape characteristics at least partly because mating with such women will produce healthier offspring. Although a prediction of this *adaptation-for-mate-choice* hypothesis is that straight men will show stronger preferences for feminized versus masculinized versions of women's faces than will gay men, only one previous study has directly tested this prediction. Here we directly replicated that study by comparing 623 gay and 3163 straight men's preferences for feminized versus masculinized versions of faces. Consistent with the adaptation-for-mate-choice hypothesis of straight men's femininity preferences, we found that straight men showed significantly stronger preferences for feminized female faces than did gay men. Consistent with previous research suggesting that gay men place a premium on masculinity in potential romantic partners, we also found that gay men showed significantly stronger preferences for masculinized versions of male faces than did straight men. Together, these findings indicate the sexual orientation contributes to individual differences in men's face preferences.

**Data Availability Statement:** The data used in this study are available at https://osf.io/cuqdz.

**Funding:** This work was funded by a European Research Council (ERC) grant (KINSHIP - 647910)

## Introduction

Many studies of straight men's face preferences have reported that straight men show strong preferences for female faces with pronounced feminine shape characteristics [1, 2]. These preferences for feminine female faces are widely assumed to at least partly reflect adaptations for mate choice [1, 3]. Specifically, straight men are thought to show strong preferences for feminized versions of women's faces because mating with such women would produce healthier or more viable offspring [1, 3]. However, evidence that women displaying more feminine facial characteristics possess traits that would cause them to produce healthier, more viable offspring is mixed. For example, while some studies have reported that women with more feminine faces report better general health, such as less frequent colds and other illnesses [e.g. 4, 5], other studies have not found significant correlations between facial femininity and measures of women's health [e.g. 6, 7]. Although these mixed results suggest the proposed link between women's actual health and facial femininity, women with more feminine facial characteristics are perceived to be healthier and as likely to be better parents [2].

awarded to LMD. The funders had no role in study design, data collection and analysis, decision to publish, or preparation of the manuscript.

**Competing interests:** The authors have declared that no competing interests exist.

If straight men show strong preferences for feminized versions of women's faces because mating with such women would produce healthier or more viable offspring, one might reasonably predict that gay men's preferences for feminized versions of women's faces would be weaker than those of straight men. Such results would complement those showing opposite-sex biases in straight participants' face preferences, which have been widely interpreted as evidence that face preferences at least partly reflect adaptations for mate choice [1, 3]. However, and perhaps surprisingly, only one study has directly tested this hypothesis. In a study of 311 gay men and 215 straight men, Glassenberg et al. [8] found that straight men did indeed show stronger preferences for feminized versus masculinized versions of women's faces than did gay men. Interestingly, Glassenberg et al. [8] also found that gay men showed stronger preferences for masculinized versions of men's faces than did straight men, consistent with the proposal that gay men place a premium on masculinity in romantic partners.

Many previously reported findings for individual differences in preferences for sexually dimorphic face shapes have not replicated well in recent large-scale studies [see, e.g., 9]. Consequently, we attempted a direct replication of Glassenberg et al. [8]. We compared 623 gay and 3163 straight men's preferences for feminized versus masculinized versions of female and male faces. While Glassenberg et al. tested both men and women, our replication study focused on men's face preferences. Following Glassenberg et al., we predicted that straight men would show stronger preferences for femininity in women's faces than gay men did and that gay men would show stronger preferences for masculinity in men's faces than straight men did.

## Methods

### Participants

Participants for the online study, which was run at faceresearch.org, were 623 men (mean age = 26.4 years, SD = 7.41 years) who reported that their preferred sex of partner was male and 3163 straight men (mean age = 26.65 years, SD = 7.52 years) who reported that their preferred sex of partner was female. No other exclusion or inclusion criteria were applied. 60% of participants reported their ethnicity as White, 20% opted not to report their ethnicity, 6% reported their ethnicity as West Asian, 5% reported their ethnicity as East Asian, and 2% reported their ethnicity as African. All other ethnicities accounted for <1% of our sample. All participants provided informed consent and all procedures were approved by the Psychology Ethics Committee (University of Glasgow).

### Stimuli

Following previous studies of individual differences in women's preferences for masculine faces [e.g., 9, we used prototype-based image transformations to objectively manipulate sexual dimorphism of 2D shape in face images. First, male and female prototype (i.e. average) faces were manufactured using established computer graphic methods that have been widely used in studies of face perception [10]. These prototypes were manufactured using face images of 20 young White male adults and 20 young White female adults, respectively. Next, 50% of the linear differences in 2D shape between symmetrized versions of the male and female prototypes were added to or subtracted from face images of 20 young White male adults and 20 young White female adults. This process created masculinized and feminized versions of the individual face images that differ in sexual dimorphism of 2D shape and that are matched in other regards. Stimuli are publicly available [11].

## Procedure

Participants were shown the 40 pairs of face images and were asked to choose the face in each pair that was more attractive. As in Glassenberg et al. [8], the specific instruction was "Which face do you consider more attractive?". Participants also indicated the strength of these preferences by choosing from the options 'slightly more attractive', 'somewhat more attractive, 'more attractive', and 'much more attractive'. The order in which pairs of faces were shown was fully randomized and the side of the screen on which any particular image was shown was also fully randomized. Responses were coded using a 0 (masculinized face judged as much more attractive than feminized face) to 7 (feminized face judged as much more attractive than masculinized face). These preference scores were centered on chance before being used in our analyses.

## Results

Analyses were conducted using R v3.6.0 [12]. Data, analysis code, and full results output are publicly available at https://osf.io/cuqdz/. First, we analyzed preference scores using a mixed effect model using lmer and lmerTest [13, 14] with the factors *sexual orientation* (effect coded so that gay = -0.5 and straight = 0.5) and *sex of face* (effect coded so that female = -0.5 and male = 0.5), and the covariate *participant age* (centered and scaled on mean of sample). We included participant age as a covariate to control for possible effects of age on face preferences [15]. Random intercepts were included for participant and stimulus, with random slopes specified maximally [16, 17]. We did not include the interaction between sexual orientation and participant age in our model.

Results of this initial analysis are summarized in Fig 1. The intercept was significant and positive (estimate = 0.29, SE = 0.06, df = 39.8, t = 4.54, p<0.001), indicating that participants generally preferred feminized versions of faces. The main effect of sexual orientation was significant and positive (estimate = 0.39, SE = 0.03, df = 88.8, t = 11.72, p<0.001), indicating that straight men generally showed stronger preferences for feminized versions of faces than did gay men (see Fig 1). The main effect of sex of face was significant and negative (estimate = -0.90, SE = 0.12, df = 39.2, t = -7.07, p<0.001), indicating that men generally showed stronger preferences for feminized versions of faces when judging the attractiveness of female than male faces (see Fig 1). The effect of participant age was not significant (estimate = 0.01, SE = 0.01, df = 226.6, t = 1.14, p = 0.252). The interaction between sex of face and sexual orientation was significant (estimate = 0.23, SE = 0.06, df = 68.1, t = 3.70 p<0.001) and is illustrated in Fig 1. Repeating this analysis without participant age as a covariate showed the same pattern of results (see robustness analysis reported at https://osf.io/cuqdz/).

Next, we repeated the analysis described above separately for male and female faces and with sex of face removed from the model. Our analysis of female faces showed a significant effect of sexual orientation (estimate = 0.28, SE = 0.03, df = 53.9, t = 7.26, p<0.001), indicating that straight men showed stronger preferences for feminized versions of women's faces than did gay men (see Fig 1). The effect of participant age was not significant (estimate = -0.01, SE = 0.01, df = 115.4, t = -0.58, p = 0.560). Our analysis of male faces showed a significant effect of sexual orientation (estimate = 0.51, SE = 0.05, df = 37.0, t = 9.89, p<0.001), indicating that gay men showed stronger preferences for masculinized versions of men's faces than did straight men (see Fig 1). The effect of participant age was significant (estimate = 0.03, SE = 0.01, df = 107.0, t = 2.36, p = 0.019) and indicated that older men showed stronger preferences for feminized male faces.

Preferences for femininity in women's faces were significantly greater than chance when gay and straight men judged the attractiveness of women's faces (both ps < .001). Preferences

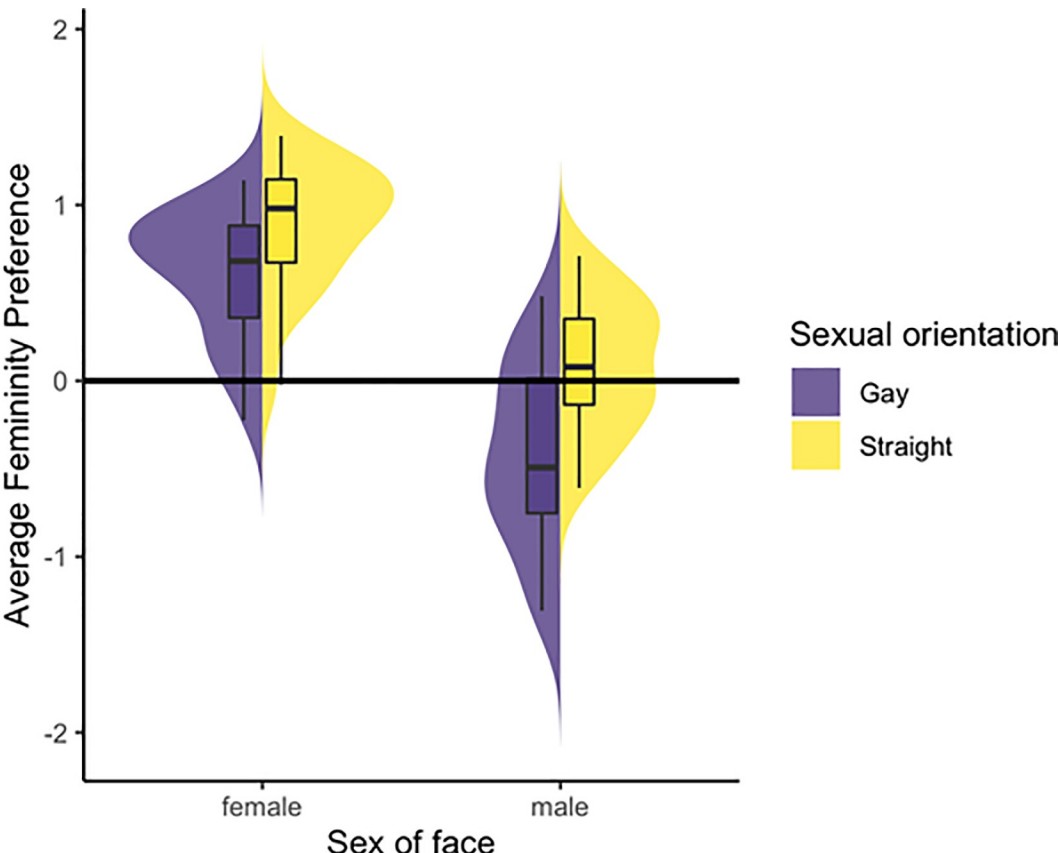

**Fig 1. The interaction between sexual orientation and sex of face on femininity preference in our study.** The box plots and distributions represent the average femininity score for each individual face. The box plots are showing the median, first and third quartile, and the minimum and maximum femininity score for gay (purple) and straight (yellow) participants. On the y-axis, zero equals chance.

for femininity in men's faces were significantly lower than chance when gay men judged the attractiveness of men's faces (p = .002), but did not differ significantly from chance when straight men judged the attractiveness of men's faces (p = .28).

## Discussion

Our analyses showed that straight men demonstrated stronger preferences for feminized versions of women's faces than did gay men. This pattern of results replicates those reported by Glassenberg et al. [8]. This pattern of results is also consistent with the proposal that straight men's strong preferences for women's faces with pronounced feminine characteristics at least partly reflect mating-related motivations [1, 3]. Since women's faces displaying feminine characteristics are generally ascribed prosocial traits [2], that both gay and straight men showed preferences for feminine version of women's faces that were significantly greater than chance suggests that men's preferences for feminine female faces might also partly reflect general preferences for prosocial associates.

Our analyses also showed that gay men demonstrated stronger preferences for masculine men that did straight men. This pattern of results also replicates those reported by Glassenberg et al. [8]. This pattern of results is also consistent with other previous research suggesting that

gay men show stronger preferences for men described in vignettes as possessing masculine traits [18]. It is currently unclear why gay men place this premium on masculinity.

A potential limitation of our study was the use of a forced-choice paradigm for assessing men's preferences for sexually dimorphic face shapes. We used this paradigm in our study because it was the same as that used in the study by Glassenberg et al. [8] that we were attempting to (and successfully did) replicate. However, some recent research suggests that forced-choice paradigms can produce qualitatively different patterns of results than other methods for assessing preferences for sexually dimorphism face-shape characteristics [19]. Establishing the extent to which the effects of sexual orientation on face preferences that we observed in the current study and that were also observed by Glassenberg et al. [8] generalize to other methods for assessing face preferences would be an important direction for future research. Since we only tested male participants, it is also not known whether Glassenberg et al's findings for women's face preferences would replicate.

In conclusion, we found that straight men showed stronger preferences for feminized versions of women's faces than did gay men, consistent with an adaptation-for-mate-choice explanation of straight men's preferences for feminine women. We also found that gay men showed stronger preferences for masculine men that did straight men. Together these results suggest that sexual orientation influences men's face preferences.

## Author Contributions

**Conceptualization:** Victor Shiramizu, Lisa M. DeBruine, Benedict C. Jones.

**Formal analysis:** Victor Shiramizu, Ciaran Docherty, Lisa M. DeBruine.

**Methodology:** Victor Shiramizu, Lisa M. DeBruine, Benedict C. Jones.

**Resources:** Lisa M. DeBruine, Benedict C. Jones.

**Software:** Lisa M. DeBruine.

**Supervision:** Lisa M. DeBruine, Benedict C. Jones.

**Writing – original draft:** Victor Shiramizu, Ciaran Docherty, Lisa M. DeBruine, Benedict C. Jones.

**Writing – review & editing:** Victor Shiramizu, Ciaran Docherty, Lisa M. DeBruine, Benedict C. Jones.

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
