## [Decision Letter · Decision Letter 0]

18 Sep 2020

PONE-D-20-06065

Sexual orientation predicts men’s preferences for sexually dimorphic face-shape characteristics

PLOS ONE

Dear Dr. Victor Shiramizu ,

Thank you for submitting your manuscript to PLOS ONE. After careful consideration, we feel that it has merit but does not fully meet PLOS ONE’s publication criteria as it currently stands. Therefore, we invite you to submit a revised version of the manuscript that addresses the points  by two reviewers.

Please submit your revised manuscript by in the of October. If you need moree time than this to complete your revisions, please reply to this message or contact the journal office at plosone@plos.org. Please include the following items when submitting your revised manuscript:

We look forward to receiving your revised manuscript.

Kind regards,

Tapio Mappes

Academic Editor

PLOS ONE

Journal Requirements:

2. We note that Figure 1 includes an image of participants  in the study. 

Reviewers' comments:

Reviewer's Responses to Questions

**Comments to the Author**

1. Is the manuscript technically sound, and do the data support the conclusions?

Reviewer #1: Partly

Reviewer #2: Yes

2. Has the statistical analysis been performed appropriately and rigorously? 

Reviewer #1: Yes

Reviewer #2: Yes

3. Have the authors made all data underlying the findings in their manuscript fully available?

Reviewer #1: Yes

Reviewer #2: Yes

4. Is the manuscript presented in an intelligible fashion and written in standard English?

Reviewer #1: Yes

Reviewer #2: Yes

5. Review Comments to the Author

Reviewer #1: Paper to be reviewed: Sexual orientation predicts men’s preferences for sexually dimorphic face-shape characteristics

1. Is the tittle informative?

I would suggest to include in the tittle the fact that the study is a replication, that way the reader will now straight away what to expect.

2. Does the Abstract clearly identify the need and relevance for this research?

The abstract does identify the need and relevance of the research presented. However, two important findings are mentioned (1: Consistent with the adaptation-for-mate-choice hypothesis of straight men’s femininity preferences, we found that straight men showed significantly stronger preferences for feminized female faces than did gay men; 2: Consistent with previous research, we also found that gay men showed significantly stronger preferences for masculinized versions of male faces than did straight men), but an explanation is suggested only for the for the first one. I would suggest the authors very briefly state why do they think gay men showed significantly stronger preferences for masculinized versions of male faces.

3. Does the Introduction identify the need and relevance for this research?

Major issues:

The Glassenberg’s study is mentioned, including its main findings, but similarly to the issue with the abstract, there is no reference on how did the authors explained gay men’s preferences for masculinized faces than straight men (e.g., “Our data suggest that face preferences of homosexual individuals reflect a system of bio- logically and socially guided preferences at least as complex as those found among heterosexual individuals”).

The Glassenberg’s study analysed preferences for similarities and differences between straight and gay males and females in reference to their preferences for a facial dimorphism (which isn’t done in the current study). If this study is a partial replication, it should be more specific about what is not being replicated from Glassenbergs. I believe the reader should be given more context about Glassenberg’s paper, and to what extent is this a replication. What is it missing and whats new?

Similarly, the readers may benefit from having more context information about the causes of men’s preferences for women’s feminine faces. What type of health has been measured in women? How has this been done and why is there no consensus?

Minor issues:

There is a typo in the second paragraph, a “THE” should be deleted.

4. Does the Methodology target the main question(s) appropriately?

Minor issues/Major issues:

I believe the methods used to test their hypothesis were appropriate. I appreciated how concise and clear was the description of their methodology, however it would be very helpful to get more details about what were participants asked in reference to the pictures shown. What kind of preference were they specifically asked about? Friends, partner? If the question was only about preferences in general, without any specification, then it should be acknowledged and perhaps the limitations of this addressed.

5. Are the Results clearly and logically presented, and are they justified by the data presented? Are the figures clear and fully described?

The results are clearly presented and are justified by the data presented. However, I was surprised to see that age was at some point included in the model. I wasn’t expecting it as there was no mention about any relationship between age and facial preferences in the introduction. I would suggest including a paragraph where all the predictions are stated and a potential explanation and references are given to the reader.

In terms of the figures, these were clear and fully described in their labels. I appreciated the inclusion if minimum and maximum femininity scores for both conditions (gay vs. straight).

6. Does the discussion justifiably respond to main questions the author(s) posed? Do the Conclusions go too far or not far enough based on the results?

I believe that the authors need to specify that this is a partial replication, as their sample only included men. In line with this, the conclusion that this study is a replication of Glassenger’s study may be going too far. The addition of more context of the extent of the replication will suffice.

In the first paragraph of the discussion, the authors say that pattern of results found “at least partly reflects mating-related motivations”, it would be very helpful to compliment this sentence with what other possible motivations it could reflect.

I would also suggest that the authors include an explanation for gay men preferring more masculine faces than straight men. As of right now, they are only stating the facts, but have not suggested why they think this is the case in their study.

7. Is the manuscript’s story cohesive and tightly reasoned throughout? If not, where does it deviate from the central argument?

The manuscript does present a cohesive story, however I believe that it would greatly benefit from the above comments, specially including a more detailed description of how it is, and how it isn’t a replication of Glassenger’s study.

I look forward to reviewing this manuscript’s next round of edits and seeing it published soon.

Reviewer #2: This study examined differences in facial preferences for faces of both genders in straight vs gay men. The methods and design are very straightforward. Straight men preferred feminine women’s faces more than gay men did. In contrast, gay men preferred masculine men’s faces more than straight men did. However, even with the straightforward results, I find that the MS is far too brief. Here is a list of issues that need to be addressed:

1. The introduction is very thin on the theoretical motivation of the paper. I am not really sure why a difference between straight and gay men is considered a “critical hypothesis”. Does the theory really predict that straight men need to show a stronger preference than gay men? To my mind, a null results wouldn’t have diminished the evolutionary account, nor does a significant difference provide strong support for the evolutionary account. I would like to suggest the authors provide more details on the theoretical motivation.

2. What is the ethnicity of the raters? Do results change after controlling for ethnicity (if ethnicity if mixed)?

3. Even though the results show a difference between straight and gay men, we do not know if there is a preference within each group of men. This information is critical for the interpretation of the results because we need to know that straight men do indeed show a preference for feminine women’s faces.

4. The discussion is equally brief. I would like to see more in-depth discussion of the results. For example, how does the effect size compare to that of the previous study?

6. PLOS authors have the option to publish the peer review history of their article (what does this mean?). If published, this will include your full peer review and any attached files.

Reviewer #1: **Yes: **Martha Lucia Borras Guevara

Reviewer #2: No

---

## [Author Response · Author response to Decision Letter 0]

9 Oct 2020

Reviewer #1: Paper to be reviewed: Sexual orientation predicts men’s preferences for sexually dimorphic face-shape characteristics

1. Is the tittle informative? I would suggest to include in the tittle the fact that the study is a replication, that way the reader will now straight away what to expect.

We have clarified this point in the title (New title: Sexual orientation predicts men’s preferences for sexually dimorphic face-shape characteristics: A replication study)

2. Does the Abstract clearly identify the need and relevance for this research? The abstract does identify the need and relevance of the research presented. However, two important findings are mentioned (1: Consistent with the adaptation-for-mate-choice hypothesis of straight men’s femininity preferences, we found that straight men showed significantly stronger preferences for feminized female faces than did gay men; 2: Consistent with previous research, we also found that gay men showed significantly stronger preferences for masculinized versions of male faces than did straight men), but an explanation is suggested only for the for the first one. I would suggest the authors very briefly state why do they think gay men showed significantly stronger preferences for masculinized versions of male faces.

We have clarified in the abstract that the results for male faces are consistent with the proposal that Consistent with previous research suggesting that gay men place a premium on masculinity in potential romantic partners, we also found that gay men showed significantly stronger preferences for masculinized versions of male faces than did straight men. 

Text added to Abstract: “Consistent with previous research suggesting that gay men place a premium on masculinity in potential romantic partners, we also found that gay men showed significantly stronger preferences for masculinized versions of male faces than did straight men. “

3. Does the Introduction identify the need and relevance for this research? Major issues: The Glassenberg’s study is mentioned, including its main findings, but similarly to the issue with the abstract, there is no reference on how did the authors explained gay men’s preferences for masculinized faces than straight men (e.g., “Our data suggest that face preferences of homosexual individuals reflect a system of bio- logically and socially guided preferences at least as complex as those found among heterosexual individuals”).

We have clarified this point in our Introduction and abstract.

Text added to Abstract: “Consistent with previous research suggesting that gay men place a premium on masculinity in potential romantic partners, we also found that gay men showed significantly stronger preferences for masculinized versions of male faces than did straight men.”

Text added to Introduction: “Interestingly, Glassenberg et al. [8] also found that gay men showed stronger preferences for masculinized versions of men’s faces than did straight men, consistent with the proposal that gay men place a premium on masculinity in romantic partners.”

The Glassenberg’s study analysed preferences for similarities and differences between straight and gay males and females in reference to their preferences for a facial dimorphism (which isn’t done in the current study). If this study is a partial replication, it should be more specific about what is not being replicated from Glassenbergs. I believe the reader should be given more context about Glassenberg’s paper, and to what extent is this a replication. What is it missing and whats new?

We have clarified in our Introduction that we have replicated the part of the Glassenberg et al. study related to male, not female, participants.

Text added to Introduction: “While Glassenberg et al. tested both men and women, our replication study focused on men’s face preferences.”

Similarly, the readers may benefit from having more context information about the causes of men’s preferences for women’s feminine faces. What type of health has been measured in women? How has this been done and why is there no consensus?

We have clarified this issue in our Introduction. 

Text added to Introduction: “For example, while some studies have reported that women with more feminine faces report better general health, such as less frequent colds and other illnesses [e.g. 4, 5], other studies have not found significant correlations between facial femininity and measures of women’s health [e.g. 6, 7]. Although these mixed results suggest the proposed link between women’s actual health and facial femininity, women with more feminine facial characteristics are perceived to be healthier and as likely to be better parents [2].”

Minor issues: There is a typo in the second paragraph, a “THE” should be deleted.

We have fixed this typo.

4. Does the Methodology target the main question(s) appropriately? Minor issues/Major issues: I believe the methods used to test their hypothesis were appropriate. I appreciated how concise and clear was the description of their methodology, however it would be very helpful to get more details about what were participants asked in reference to the pictures shown. What kind of preference were they specifically asked about? Friends, partner? If the question was only about preferences in general, without any specification, then it should be acknowledged and perhaps the limitations of this addressed.

We have clarified that participants were instructed to click on the face that was more attractive.

Text added to Methods: “Participants were shown the 40 pairs of face images and were asked to choose the face in each pair that was more attractive. As in Glassenberg et al. [8], the specific instruction was “Which face do you consider more attractive?”. Participants also indicated the strength of these preferences by choosing from the options ‘slightly more attractive’, ‘somewhat more attractive, ‘more attractive’, and ‘much more attractive’.”

We disagree that is would be appropriate to consider this question a limitation because asking participants to choose the more attractive face for specific contexts would necessarily confound instruction context, sex of face, and participant sexual orientation.

5. Are the Results clearly and logically presented, and are they justified by the data presented? Are the figures clear and fully described? The results are clearly presented and are justified by the data presented. However, I was surprised to see that age was at some point included in the model. I wasn’t expecting it as there was no mention about any relationship between age and facial preferences in the introduction. I would suggest including a paragraph where all the predictions are stated and a potential explanation and references are given to the reader.

We now note that age was included as a covariant because it has been found to relate to men’s face preferences in some previous work [15] and added a robustness analysis showing that the interaction between sex of face and sexual orientation is significant when age is not included in the model.

We have also clarified our predictions in our Introduction (“Following Glassenberg et al., we predicted that straight men would show stronger preferences for femininity in women’s faces than gay men did and that gay men would show stronger preferences for masculinity in men’s faces than straight men did.”)

In terms of the figures, these were clear and fully described in their labels. I appreciated the inclusion if minimum and maximum femininity scores for both conditions (gay vs. straight).

6. Does the discussion justifiably respond to main questions the author(s) posed? Do the Conclusions go too far or not far enough based on the results? I believe that the authors need to specify that this is a partial replication, as their sample only included men. In line with this, the conclusion that this study is a replication of Glassenger’s study may be going too far. The addition of more context of the extent of the replication will suffice.

We have clarified this point in the Discussion and Introduction.

Text added to Introduction: “While Glassenberg et al. tested both men and women, our replication study focused on men’s face preferences.”

Text added to Discussion: “Since we only tested male participants, it is also not known whether Glassenberg et al’s findings for women’s face preferences would replicate.”

In the first paragraph of the discussion, the authors say that pattern of results found “at least partly reflects mating-related motivations”, it would be very helpful to compliment this sentence with what other possible motivations it could reflect.

We have commented on this in our Discussion.

Text added to discussion: “Since women’s faces displaying feminine characteristics are generally ascribed prosocial traits [2], that both gay and straight men showed preferences for feminine version of women’s faces that were significantly greater than chance suggests that men’s preferences for feminine female faces might also partly reflect general preferences for prosocial associates.”

I would also suggest that the authors include an explanation for gay men preferring more masculine faces than straight men. As of right now, they are only stating the facts, but have not suggested why they think this is the case in their study.

We have clarified that it is not clear why gay men place a premium on masculinity. 

Text added to Discussion: “Our analyses also showed that gay men demonstrated stronger preferences for masculine men that did straight men. This pattern of results also replicates those reported by Glassenberg et al. [8]. This pattern of results is also consistent with other previous research suggesting that gay men show stronger preferences for men described in vignettes as possessing masculine traits [17]. It is currently unclear why gay men place this premium on masculinity.”

7. Is the manuscript’s story cohesive and tightly reasoned throughout? If not, where does it deviate from the central argument? The manuscript does present a cohesive story, however I believe that it would greatly benefit from the above comments, specially including a more detailed description of how it is, and how it isn’t a replication of Glassenger’s study. I look forward to reviewing this manuscript’s next round of edits and seeing it published soon.

Reviewer #2: This study examined differences in facial preferences for faces of both genders in straight vs gay men. The methods and design are very straightforward. Straight men preferred feminine women’s faces more than gay men did. In contrast, gay men preferred masculine men’s faces more than straight men did. However, even with the straightforward results, I find that the MS is far too brief. Here is a list of issues that need to be addressed: 1. The introduction is very thin on the theoretical motivation of the paper. I am not really sure why a difference between straight and gay men is considered a “critical hypothesis”. Does the theory really predict that straight men need to show a stronger preference than gay men? To my mind, a null results wouldn’t have diminished the evolutionary account, nor does a significant difference provide strong support for the evolutionary account. I would like to suggest the authors provide more details on the theoretical motivation.

We have clarified in our Introduction that this pattern of results would complement previous research on straight individuals that have interpreted opposite sex biases in face preferences as evidence for adaptation to mate choice.

Text added to Introduction: “If straight men show strong preferences for feminized versions of women’s faces because mating with such women would produce healthier or more viable offspring, one might reasonably predict that gay men’s preferences for feminized versions of women’s faces would be weaker than those of straight men. Such results would complement those showing opposite-sex biases in straight participants’ face preferences, which have been widely interpreted as evidence that face preferences at least partly reflect adaptations for mate choice [1,3].”

2. What is the ethnicity of the raters? Do results change after controlling for ethnicity (if ethnicity if mixed)? 

We now report in our Methods that the majority of our participants are white.

Text added to Methods: 60% of participants reported their ethnicity as White, 20% opted not to report their ethnicity, 6% reported their ethnicity as West Asian, 5% reported their ethnicity as East Asian, and 2% reported their ethnicity as African. All other ethnicities accounted for <1% of our sample.

3. Even though the results show a difference between straight and gay men, we do not know if there is a preference within each group of men. This information is critical for the interpretation of the results because we need to know that straight men do indeed show a preference for feminine women’s faces. 

We have clarified that these preferences were significantly different from chance in our Results: “Preferences for femininity in women’s faces were significantly greater than chance when gay and straight men judged the attractiveness of women’s faces (both ps < .001). Preferences for femininity in men’s faces were significantly lower than chance when gay men judged the attractiveness of men’s faces (p = .002), but did not differ significantly from chance when straight men judged the attractiveness of men’s faces (p = .28).”

Text added to Results: “Repeating this analysis without participant age as a covariate showed the same pattern of results (see robustness analysis reported at https://osf.io/cuqdz/).”

4. The discussion is equally brief. I would like to see more in-depth discussion of the results. For example, how does the effect size compare to that of the previous study?

Comparing effect sizes between the two studies is not straightforward since Glassenberg et al. aggregated preferences across trials, whereas our LME model takes into account variation across trials.

---

## [Editor Report · Decision Letter 1]

30 Oct 2020

Sexual orientation predicts men’s preferences for sexually dimorphic face-shape characteristics: a replication study.

PONE-D-20-06065R1

Dear Dr. Victor Shiramizu,

We’re pleased to inform you that your manuscript has been judged scientifically suitable for publication and will be formally accepted for publication once it meets all outstanding technical requirements.

Kind regards,

Tapio Mappes

Academic Editor

PLOS ONE
---

## [Editor Report · Acceptance letter]

5 Nov 2020

PONE-D-20-06065R1 

Sexual orientation predicts men’s preferences for sexually dimorphic face-shape characteristics: A replication study 

Dear Dr. Shiramizu:

I'm pleased to inform you that your manuscript has been deemed suitable for publication in PLOS ONE. Congratulations! Your manuscript is now with our production department. 

Kind regards, 

on behalf of

Prof Tapio Mappes 

Academic Editor

PLOS ONE